# Quantifying Sub-Elite Youth Football Weekly Training Load and Recovery Variation

**José E. Teixeira** [1,2,3,*], **Pedro Forte** [1,3,4], **Ricardo Ferraz** [1,5], **Miguel Leal** [4], **Joana Ribeiro** [4], **António J. Silva** [1,2], **Tiago M. Barbosa** [1,3] **and António M. Monteiro** [1,3]

[1] Research Centre in Sports Sciences, Health and Human Development, 5001-801 Vila Real, Portugal; pedromiguel.forte@iscedouro.pt (P.F.); rmpf@ubi.pt (R.F.); ajsilva@utad.pt (A.J.S.); barbosa@ipb.pt (T.M.B.); mmonteiro@ipb.pt (A.M.M.)
[2] Department of Sports, Exercise and Health Sciences, University of Trás-os-Montes e Alto Douro, 5001-801 Vila Real, Portugal
[3] Departamento de Desporto e Educação Física, Instituto Politécnico de Bragança, 5300-253 Bragança, Portugal
[4] Department of Sports, Douro Higher Institute of Educational Sciences, 4560-708 Penafiel, Portugal; amnfla@gmail.com (M.L.); joana.ribeiro@iscedouro.pt (J.R.)
[5] Department of Sports Sciences, University of Beira Interior, 6201-001 Covilhã, Portugal
* Correspondence: jose.eduardo@ipb.pt

**Abstract:** Monitoring the training load in football is an important strategy to improve athletic performance and an effective training periodization. The aim of this study was two-fold: (1) to quantify the weekly training load and recovery status variations performed by under-15, under-17 and under-19 sub-elite young football players; and (2) to analyze the influence of age, training day, weekly microcycle, training and playing position on the training load and recovery status. Twenty under-15, twenty under-17 and twenty under-19 players were monitored over a 2-week period during the first month of the 2019–2020 competitive season. Global positioning system technology (GPS) was used to collect external training loads: total distance covered, average speed, maximal running speed, relative high-speed running distance, high metabolic load distance, sprinting distance, dynamic stress load, accelerations and decelerations. Internal training load was monitored using ratings of perceived exertion (RPE) and session rating of perceived exertion (sRPE). Recovery status was obtained using the total quality recovery (TQR) scale. The results show an age-related influence for external training load ($p \leq 0.001$; $d = 0.29$–$0.86$; moderate to strong effect), internal training load ($p \leq 0.001$, $d = 0.12$–$0.69$; minimum to strong effect) and recovery status ($p \leq 0.001$, $d = 0.59$; strong effect). The external training load presented differences between training days ($p < 0.05$, $d = 0.26$–$0.95$; moderate to strong effect). The playing position had a minimum effect on the weekly training load ($p < 0.05$; $d = 0.06$–$0.18$). The weekly microcycle had a moderate effect in the TD ($p < 0.05$, $d = 0.39$), RPE ($p < 0.05$; $d = 0.35$) and sRPE ($p < 0.05$, $d = 0.35$). Interaction effects were found between the four factors analyzed for deceleration ($F = 2.819$, $p = 0.017$) and between inter-day, inter-week and age for total covered distance ($F = 8.342$, $p = 0.008$). This study provided specific insights about sub-elite youth football training load and recovery status to monitor training environments and load variations. Future research should include a longer monitoring period to assess training load and recovery variations across different season phases.

**Keywords:** monitoring; workload; perceived exertion; soccer; periodization





## 1. Introduction

Monitoring training load in football is an important strategy to improve athletic performance and effective training periodization [1,2]. Determining individual adaptations allows the assessment of physical and physiological responses, gathering insights about fatigue-recovery status [3,4]. In addition, there is evidence that an optimal load management can minimize the risk for overtraining and injuries [5].

The training load has been defined as an input variable for training outcomes [6]. It is possible to slip it into external and internal loads [7]. The external load describes the performed work, whereas the internal load refers to biological requirements (physiological and psychological) imposed on athletes [8]. The external load can be monitored by global positioning system (GPS) devices and micro-electromechanical systems (MEMS) [9]. These tracking systems can accurately measure distances, speeds, accelerations/decelerations and accelerometer variables (e.g., player load, body impact or dynamic stress load) [5,10,11]. The internal load is assessed using objective and subjective measures [12]. The rating of perceived exertion (RPE) is the most common scale used to assess individual internal load [13]. Validity and reliability of RPE are well established in several sports and physical activities at different ages and expertise levels [14]. The session rating of perceived exertion (sRPE) has been reported as a better training load indicator to explain the training session intensity in comparison to RPE [15].

An adequate balance between stress and recovery has been described as essential to optimize athletic performance [6–8]. Kenttä and Hassmén [16] proposed "total quality recovery" (TQR) to predict individual and subjective recovery [17]. The TQR scale describes the fatigue-recovery using physical and psychosocial components [17,18]. Studies have verified that the TQR did not contribute to predict young football performance. Nevertheless, the hypothesis was partly confirmed when only the weekly training load was analyzed [18].

The training load quantification has been widely studied in professional football for training conditions [19–24]. The training load presented a high load variation within a weekly microcycle [21,23,25]. In contrast, the cumulative training load presented limited variation along mesocycles [20,22,26]. A large variation in training load has been also documented concerning training modes [19,22]. The positional role should be considered an important variable to quantify the training load, given the interposition differences [20,23,24]. Generally, central midfielders covered more distance and wide-midfielders covered more distances in high-intensity zones [20,23,24]. The central defenders and wide-defenders covered more distance in low-intensity zones. In addition, forwards seemed to sprint significantly less frequently than central defenders [23].

In youth football, the influence of age and positional role on the activity profile has been analyzed in the match behavior [27,28] and constrained training tasks [29,30]. Typical weekly training load has been also analyzed in elite young football players; there was an age-related influence on the total weekly training load, which seems to be increasing with age [31–33]. For instance, Abade et al. [32] reported higher total distances covered in under-17 (U17), followed by under-19 (U19) and under-15 (U15) players. In the same study, the total body impacts and relative impacts were lower in U15 players. Coutinho et al. [31] reported a higher total distance covered (<13 km h$^{-1}$), body impacts and time spent above high-intensity zones in U19 post-match training sessions. Building upon that, the elite typical weekly training load presented significant differences within weekly microcycles and among training modes [31–33].

To the best of our knowledge, there is a lack of literature about training load and recovery variations across a competitive season in sub-elite youth football. More specifically, there are few studies in sub-elite youth football, concerning the research topic of training load and recovery. Building upon that, training load variations in sub-elite players may depend on socio-economics, demographics, experience and competitive levels [3–10]. As far as we are concerned, there has not been any study assessing training load and recovery status in Portuguese sub-elite football players yet. Additionally, previous studies were conducted in elite youth football academies; wherefore, this is the first study in a sub-elite academy. Another research gap is the recovery status assessment. There is only one study addressing a recovery scale in youth football and it is important to understand the recovery variations across a competitive season [18]. Moreover, this study will allow us to determine if the sub-elite players have significant training load variations by age groups, weekly microcycles, training days and positional roles.

Based on the above-mentioned information, the aims of this study were to: (i) to quantify the weekly training load and recovery variations performed by U15, U17 and U19 young players in a sub-elite football academy; and (ii) to analyze the influence of age, training day, weekly microcycle, training and playing position on the training load and recovery status. It was hypothesized that there are significant differences in weekly training load and recovery variations according to age group, weekly microcycle, training day and playing position.

## 2. Materials and Methods

### 2.1. Participants

Sixty male football players were monitored during a 2-week period in a sub-elite Portuguese football academy: twenty U15 players (age: 13.2 ± 0.5 y; height: 1.69 ± 0.78 m; weight: 55.7 ± 9.4 kg), twenty U17 players (age: 15.4 ± 0.5 ± 1.2 y; height: 1.8 ± 0.5 m; weight: 64.38 ± 6.6 kg) and twenty U19 players (age: 17.39 ± 0.55 ± 1.8 ± 0.7 y; height: 1.82 ± 0.01 m; weight: 68.9 ± 8.4 kg). The daily training load was continuously monitored in the three age groups during the first month of the 2019–2020 competitive season. All participants were notified about the aims and risks of the investigation. The study includes only players that have signed the informed consent. The present research was conducted according to the ethical standards of the Declaration of Helsinki. The experimental approach was approved and followed by the local Ethical Committee from University of Trás-os-Montes e Alto Douro (3379-5002PA67807).

### 2.2. Experimental Approach

The training data corresponded to a total of 18 training sessions and 324 observation cases. The eligibility criteria for individual data sets considered a competitive 1-game week schedule and complete full training sessions. The microcycle included 3 training sessions per week (~90 min). The match data were not included in the analysis. The training days were classified as "match day minus format" (MD): MD-3 (Tuesday), MD-2 (Wednesday) and MD-1 (Friday).

The sampled players were characterized based on one out of five playing positions: central defenders (CD), fullbacks (FB), central midfielders (CM), wide midfielders (WM) and forwards (FW). The number of observations per position role was: CD ($n$ = 79); FB ($n$ = 65); CM ($n$ = 70); WM ($n$ = 62); and FW ($n$ = 48). The goalkeeper participated in the training session but was excluded in the present training load analysis. The training sessions had, on average, 18 players. All age groups were trained on an outdoor pitch with official dimensions (FIFA standard; 100 × 70 m). The training sessions were performed on synthetic turf pitches, from 10:00 a.m. to 08:00 p.m. and with similar environment conditions (14–20 °C; relative humidity 52–66%).

The sampled training sessions were categorized according to specific focus, following the discussion with the coach staff. All sampled training sessions started with a standard warm-up with low-intensity running, dynamic stretching for main locomotive lower limb muscles, technical actions and ball possession. The weekly training overview presented as potentially variable between categories, as different training modes with emphasis on the game-based situations and sport-specific skills for football-specific exercises [32,34].

### 2.3. Methodology

The outfields players were monitored, resorting to a portable GPS throughout the whole training session (STATSports Apex®, Newry, Northern Ireland). The GPS device provides raw position velocity and distance at 18 Hz sampling frequencies, including an accelerometer (100 Hz), a magnetometer (10 Hz) and a gyroscope (100 Hz). Each player kept this micro-technology inside a mini pocket of a custom-made vest supplied by the manufacturer, which was placed on the upper back between both scapulae. All devices were activated 30 min before the training data collection to allow for an acceptable clear reception of the satellite signal. Concerning the optimal signal for the measurement

of human movement, the match data considered eight available satellite signals as the minimum for the observations [35]. The validity and reliability of the global navigation satellite systems (GNSS), such as the GPS tracking, have been well established in the literature [36–38]. The current variables and thresholds should consider a small error of around 1–2% reported in the 10 Hz STATSports Apex® units [36].

### 2.4. Training Load Measures

#### 2.4.1. External Training Load

The external training loads were obtained with time–motion data: total distance (TD) covered (m), average speed (AvS), maximal running speed (MRS) (m·s$^{-1}$), relative high-speed running (rHSR) distance (m), high metabolic load distance (HMLD) (m), sprinting (SPR) distance (m), dynamic stress load (DSL) (a.u.), number of accelerations (ACC) and number of decelerations (DEC). The GPS software provided information only about the locomotor categories above 19.8 km·h$^{-1}$: rHSR (19.8–25.1 km·h$^{-1}$) and SPR (>25.1 km·h$^{-1}$). The sprints were measured by number and average sprint distance (m). The HMLD is a metabolic variable defined as the distance, expressed in meters, covered by a player when the metabolic power exceeds 25.5 W·kg$^{-1}$. HMLD variables include all high-speed running, accelerations and decelerations above 3 m·s$^{-2}$ [39]. Both acceleration variables (ACC/DEC) considered the movements made in the maximum intensity zone (>3 m·s$^{-2}$ and <3 m·s$^{-2}$, respectively). DSL variables were evaluated by a 100 Hz triaxial accelerometer integrated into the GPS devices. The sum of the accelerations in the three orthogonal axes of movement (X, Y and Z planes) comprise the following magnitude vector (expressed as a G force):

$$\left(\left(a_{y1} - a_{y-1}\right)^2 + a_{x1} - a_{x-1}\right)^2 + \left(a_{z1} - a_{z-1}\right)^2)$$

where $a_x$ = medio-lateral acceleration, $a_y$ = anteroposterior acceleration and $a_z$ = vertical acceleration. The DSL was expressed in arbitrary units (a.u.) [40].

The high-intensity activity thresholds were adapted from previous studies [41,42]. The GPS variables were recorded for each individual player. Training data were excluded from the analysis in the case of data collection errors, injury events, missing training or early withdrawal. The exclusion criteria resulted in the elimination of 36 observation cases.

#### 2.4.2. Internal Training Load

The RPE scale proposed by Foster et al. [1] modified the Borg's Category Ratio-10 (CR-10) to monitor exercise. Daily total training load was calculated with the sum of accumulated training load [43]. In football training the validity has been well-established previously through the correlations between changes in RPE and heart rates measures [44–46]. RPE rating was collected individually at approximately 30 min after each training session using a Microsoft Excel® spreadsheet. Players were already familiarized with the process of reporting RPE for weeks prior to data collection. The sRPE was obtained by multiplying total duration of training sessions for each individual RPE score (sRPE = RPE × session duration) following a scale from 6 to 20 [18].

#### 2.4.3. Recovery Status

To monitor recovery, each player was asked to report the TQR score on a scale from 6 to 20. This scale was proposed by Kenttä and Hassmén [16] to measure athletes' recovery perceptions. Previous research integrated the TQR score examining perceived stress and fatigue in youth football [4,47,48]. In our research, the application of the TQR scale was preceded by familiarization some weeks before data collection. The TQR was applied individually at approximately 30 min before each training session using a Microsoft Excel® spreadsheet.

### 2.5. Statistical Analysis

Descriptive statistics, the Kolmogorov–Smirnov and Levene's test were used to assess the normality and homogeneity. A one-way analysis of variance (ANOVA) for repeated measures were tested to identify differences between age group, playing position and training day. A factorial ANOVA (factor 1: age group, factor 2: training day, factor 3: weekly microcycle, factor 4: playing position) was used to analyze changes in the external training load, sRPE, RPE and TQR during training sessions. When a significant difference occurred, Bonferroni post-hoc tests were used to identify localized effects. Games–Howell post-hoc tests were applied if variances were not homogeneous. The effect size index (eta square: $\eta^2$) was computed and interpreted as: (i) without effect if $0 < \eta^2 \leq 0.04$; (ii) minimum if $0.04 < \eta^2 \leq 0.25$; (iii) moderate if $0.25 < \eta^2 \leq 0.64$; and (iv) strong if $\eta^2 > 0.64$ [49]. The independent t-test was performed to analyze the differences between weekly microcycles. Standardized effect sizes (ES) were calculated with Cohen's *d*, classified as: without effect if $d < 0.2$, moderate effect if $0.2 > d \geq 0.5$ and strong effect if $d > 0.5$ [50,51]. Statistical significance was set at $p < 0.05$. Data are presented as the mean $\pm$ one standard deviation (SD). Mean differences ($\Delta$) were presented in absolute values and percentage (%). All statistical analyses were conducted using SPSS for Windows Version 22.0 (SPSS Inc., Chicago, IL, USA).

## 3. Results

### 3.1. Age Group Analysis

Table 1 presents the descriptive statistics for mean weekly training load and recovery status for each age group examined. The age groups presented significant differences in all external training load measures, except DSL. The internal training load measures presented a significant difference between age group comparison for RPE and sRPE. The TQR score presented significant differences within age groups.

**Table 1.** Mean training load and recovery status per session for each age group examined.

|  | Variables | U15 ($n$ = 102) | U17 ($n$ = 99) | U19 ($n$ = 120) | $F$ | $p$ | $\eta^2$ | Post-Hoc |
|---|---|---|---|---|---|---|---|---|
| **External load** | TD (m) | 5316.18 ± 1354.45 | 6021.45 ± 1675.64 | 4750.43 ± 1593.46 | 18.465 | 0.000 | 0.103 | a,b,c |
|  | AvS (m·min$^{-1}$) | 49.96 ± 16.35 | 56.84 ± 34.51 | 45.83 ± 15.60 | 6.192 | 0.002 | 0.037 | a |
|  | MRS (m·s$^{-1}$) | 6.58 ± 0.82 | 7.94 ± 3.12 | 7.43 ± 1.15 | 13.014 | 0.000 | 0.075 | a,b |
|  | rHSR (m) | 53.23 ± 58.34 | 166.06 ± 458.95 | 72.41± 65.95 | 5.525 | 0.004 | 0.033 | a,c |
|  | HMLD (m) | 489.11 ± 228.44 | 730.56 ± 483.38 | 524.90 ± 291.37 | 14.395 | 0.000 | 0.082 | a,c |
|  | Average sprint (m) | 28.13 ± 41.66 | 130.42 ± 462.56 | 40.16 ± 50.43 | 4.773 | 0.009 | 0.029 | a,c |
|  | Number of sprints | 1.85 ± 2.46 | 4.83 ± 4.81 | 3.12 ± 2.92 | 18.363 | 0.000 | 0.103 | a,b,c |
|  | DSL (a.u.) | 247.21 ± 135.86 | 261.28 ± 121.73 | 245.19 ± 144.87 | 0.439 | 0.645 | 0.003 | - |
|  | ACC (m·s$^{-2}$) | 33.62 ± 18.80 | 53.76 ± 20.62 | 49.90 ± 20.19 | 26.636 | 0.000 | 0.156 | a,b |
|  | DEC (m·s$^{-2}$) | 30.27 ± 19.77 | 49.77 ± 25.08 | 44.01 ± 22.53 | 20.103 | 0.000 | 0.111 | a,b |
| **Internal load** | RPE (a.u.) | 13.73 ± 1.91 | 13.51 ± 1.76 | 12.45 ± 2.50 | 11.964 | 0.000 | 0.069 | a,c |
|  | sRPE (a.u.) | 1235.29 ± 171.87 | 1215.46 ± 158.71 | 1120.24 ± 224.69 | 11.964 | 0.000 | 0.069 | a,c |
| **Recovery status** | TQR (a.u.) | 16.38 ± 1.92 | 16.24 ± 1.81 | 15.21 ± 2.16 | 11.923 | 0.000 | 0.103 | a,c |

Significant differences are verified as: (a) U15 vs. U17; (b) U15 vs. U19; (c) U17 vs. U19. Abbreviations: ACC—acceleration; a.u.—arbitrary unit; AvS—average speed; DEC—deceleration; *F*—*F* statistic; HMLD—high metabolic load distance; m—meters; min—minutes; MRS—maximum running speed; *p*—*p* value; RPE—ratings of perceived exertion; s—seconds; sRPE—session ratings of perceived exertion; TD—total distance; TQR—total quality recovery; U—under; $\eta^2$—eta-squared.

Age group differences were found for TD ($p < 0.001$, $\Delta = 705.27–1271.02$ m, $d = 0.46–0.78$), AvS ($p = 0.002$, $\Delta = 11.01$ m·min$^{-1}$, $d = 0.41$), MSR ($p < 0.001$, $\Delta = 0.86–1.36$ m·s$^{-1}$, $d = 0.60$), rHSR ($p < 0.05$, $\Delta = 93.65–112.83$ m, $d = 0.29–0.35$), HMLD ($p < 0.001$, $\Delta = 93.65–241.45$ m, $d = 0.29$), average sprint distance ($p < 0.05$, $\Delta = 90.26–102.30$ m, $d = 0.31$), number of sprints ($p < 0.05$, $\Delta = 1.72–2.97$, $d = 0.29–0.43$), ACC ($p < 0.001$, $\Delta = 16.29–20.14$ m·s$^{-2}$, $d = 0.86–1.02$) and DEC ($p < 0.001$, $\Delta = 13.74–19.50$ m·s$^{-2}$, $d = 0.29$).

The RPE and sRPE were statistically significant between age groups ($p < 0.001$, $\Delta = 1.28\text{–}1.58$ a.u., $d = 0.12\text{–}0.58$). The TQR score presented significant differences within age groups ($p = 0.000$, $F = 11.2$, $d = 0.52\text{–}0.58$). Figure 1 presents the mean differences (%) between each age group examined for external load, sRPE, RPE and TQR.

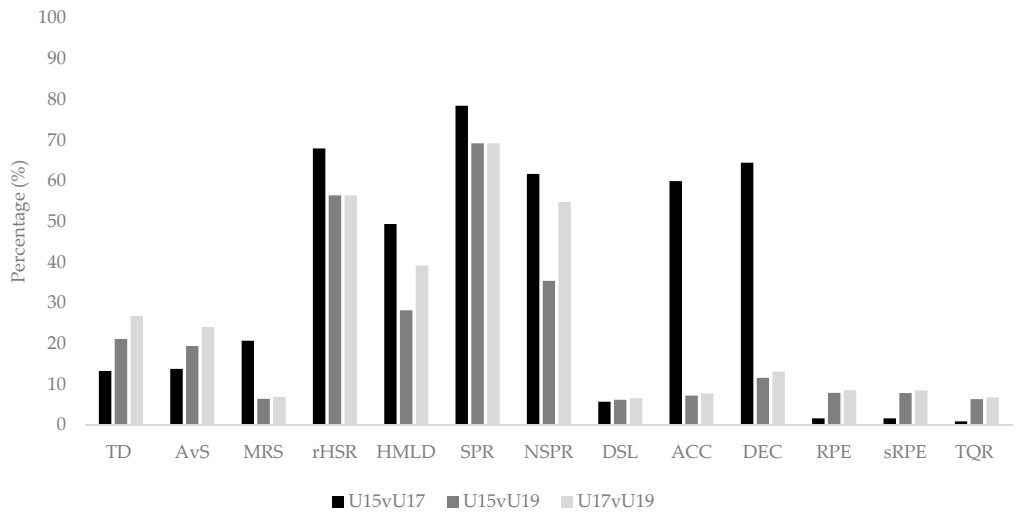

**Figure 1.** Mean training load and recovery status per session according to comparison between age groups.

### 3.2. Inter-Day Analysis

Table 2 presents the descriptive statistics for mean weekly training load and recovery status for each training day. The inter-day analysis presented significant differences in the TD covered, MRS, AvS, rHSR, HMLD, average sprint distance, DSL and DEC. The RPE, sRPE and TQR score did not present significant differences in the inter-day analysis.

**Table 2.** Mean training load and recovery status per session for each training day examined.

| | Variables | MD-3 ($n = 41$) | MD-2 ($n = 38$) | MD-1 ($n = 44$) | $F$ | $p$ | $\eta^2$ | Post-Hoc |
|---|---|---|---|---|---|---|---|---|
| **External load** | TD (m) | $5372.00 \pm 1452.14$ | $5795.99 \pm 1773.31$ | $4728.01 \pm 1618.62$ | 9.90 | 0.000 | 0.058 | a,b |
| | AvS (m·min$^{-1}$) | $53.11 \pm 17.90$ | $44.64 \pm 13.71$ | $51.82 \pm 36.42$ | 3.80 | 0.023 | 0.007 | a |
| | MRS (m·s$^{-1}$) | $7.50 \pm 2.17$ | $6.81 \pm 1.00$ | $7.52 \pm 2.33$ | 3.90 | 0.021 | 0.024 | a,c |
| | rHSR (m) | $75.42 \pm 63.06$ | $68.45 \pm 73.08$ | $87.64 \pm 102.71$ | 3.29 | 0.001 | 0.001 | - |
| | HMLD (m) | $591.17 \pm 284.94$ | $568.24 \pm 287.70$ | $488.79 \pm 293.58$ | 3.52 | 0.008 | 0.002 | - |
| | Average sprint (m) | $39.71 \pm 49.09$ | $40.40 \pm 51.11$ | $58.09 \pm 76.46$ | 3.90 | 0.048 | 0.024 | a |
| | Number of sprints | $3.13 \pm 2.94$ | $2.90 \pm 3.71$ | $3.80 \pm 4.68$ | 1.45 | 0.237 | 0.009 | - |
| | DSL (a.u.) | $267.55 \pm 144.38$ | $252.17 \pm 127.67$ | $219.21 \pm 120.30$ | 3.55 | 0.030 | 0.022 | b |
| | ACC (m·s$^{-2}$) | $48.85 \pm 22.83$ | $43.58 \pm 20.54$ | $43.21 \pm 19.87$ | 2.61 | 0.075 | 0.016 | - |
| | DEC (m·s$^{-2}$) | $45.99 \pm 25.58$ | $40.33 \pm 20.80$ | $34.44 \pm 21.81$ | 10.65 | 0.001 | 0.041 | b |
| **Internal load** | RPE (a.u.) | $13.29 \pm 2.35$ | $12.51 \pm 1.74$ | $13.27 \pm 2.28$ | 1.12 | 0.328 | 0.007 | - |
| | sRPE (a.u.) | $1196.05 \pm 211.17$ | $1158.05 \pm 211.17$ | $1194.35 \pm 205.23$ | 1.12 | 0.328 | 0.007 | - |
| **Recovery status** | TQR (a.u.) | $15.99 \pm 2.26$ | $15.82 \pm 1.76$ | $15.81 \pm 1.95$ | 0.10 | 0.907 | 0.002 | - |

Significant differences are verified as: (a) MD-1 vs. MD-2; (b) MD-1 vs. MD-3; (c) MD-2 vs. MD-3; Abbreviations: ACC—acceleration; a.u.—arbitrary unit; AvS—average speed; DEC—deceleration; $F$—$F$ statistic; HMLD—high metabolic load distance; m—meters; MD—match day; min—minutes; MRS—maximum running speed; $p$—$p$ value; RPE—ratings of perceived exertion; s—seconds; sRPE—session ratings of perceived exertion; TD—total distance; TQR—total quality recovery; $\eta^2$—eta-squared.

Post-hoc analysis found a significantly lower TD in MD-1 than MD-2 ($p = 0.009$, $\Delta = 643.99$ m, $d = 0.26$) and MD-3 ($p = 0.000$, $\Delta = 1067.98$ m, $d = 3.04$). MD-2 presented a significantly higher AvS than MD-1 ($p = 0.023$, $\Delta = 8.46$ m·min$^{-1}$, $d = 0.63$). The MRS was lower in MD-2 than MD-1 ($p = 0.032$, $\Delta = 0.68$ m·s$^{-1}$, $d = 0.43$) and MD-3 ($p = 0.000$,

$\Delta = 0.71$ m·s$^{-1}$, $d = 0.95$). MD-3 presented a significantly higher DSL following MD-1 ($p = 0.024$, MD = 48.34 a.u., $d = 0.47$). No significant differences were found between days for rHSR and HMLD. Figure 2 presents the mean differences (%) between each training day examined for external load, sRPE, RPE and TQR.

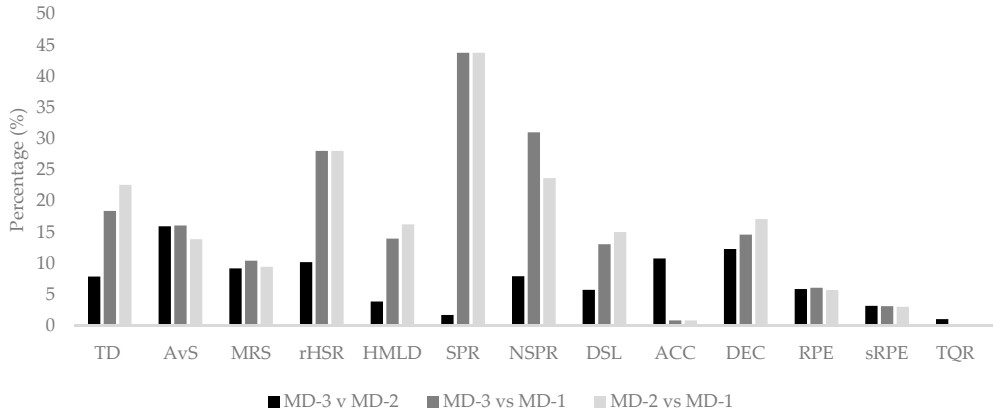

**Figure 2.** Mean training load and recovery status per session according to comparison between training days.

### 3.3. Inter-Week Analysis

Table 3 presents the descriptive statistics for mean weekly training load and recovery status for each weekly microcycle.

**Table 3.** Mean training load and recovery status per session for each week examined.

|  | Variables | Wk1 ($n = 60$) | Wk2 ($n = 42$) | Total ($n = 122$) | $F$ | $p$ | $d$ |
|---|---|---|---|---|---|---|---|
| **External load** | TD (m) | 5700.74 ± 1356.59 | 4766.80 ± 1159.89 | 5316.90 ± 1630.62 | 5.323 | 0.022 | 0.39 |
| | AvS (m·min$^{-1}$) | 48.97 ± 19.19 | 51.38 ± 11.19 | 50.49 ± 23.61 | 12.404 | 0.000 | 0.18 |
| | MRS (m·s$^{-1}$) | 6.53 ± 0.81 | 6.65 ± 0.84 | 7.32 ± 1.99 | 2.777 | 0.097 | 0.23 |
| | rHSR (m) | 51.23 ± 60.87 | 56.07 ± 55.11 | 94.98 ± 262.50 | 0.118 | 0.732 | 0.06 |
| | HMLD (m) | 515.93 ± 216.44 | 450.79 ± 242.05 | 576.47 ± 360.56 | 0.380 | 0.538 | 0.20 |
| | Average sprint (m) | 26.20 ± 41.56 | 30.88 ± 42.16 | 50.49 ± 23.61 | 0.847 | 0.358 | 0.06 |
| | Number of sprints | 1.73 ± 2.02 | 2.02 ± 2.60 | 3.24 ± 3.68 | 0.136 | 0.712 | 0.16 |
| | DSL (a.u.) | 249.90 ± 134.94 | 243.36 ± 138.71 | 250.74 ± 135.07 | 0.524 | 0.470 | 0.003 |
| | ACC (m·s$^{-2}$) | 36.35 ± 18.85 | 29.71 ± 18.25 | 45.95 ± 21.59 | 1.765 | 0.185 | 0.22 |
| | DEC (m·s$^{-2}$) | 30.60 ± 17.69 | 29.79 ± 22.62 | 41.44 ± 23.83 | 1.523 | 0.218 | 0.31 |
| **Internal load** | RPE (a.u.) | 12.83 ± 2.20 | 13.59 ± 2.10 | 13.17 ± 2.18 | 0.447 | 0.002 | 0.35 |
| | sRPE (a.u.) | 1154.75 ± 197.66 | 1222.65 ± 189.27 | 1185.56 ± 196.54 | 0.447 | 0.002 | 0.35 |
| **Recovery status** | TQR (a.u.) | 15.84 ± 2.17 | 15.96 ± 1.89 | 15.90 ± 2.05 | 3.079 | 0.608 | 0.06 |

Significant differences are verified between weeks. Abbreviations: ACC—acceleration; a.u.—arbitrary unit; AvS—average speed; DEC—deceleration; $F$—$F$ statistic; HMLD—high metabolic load distance; m—meters; min—minutes; MRS—maximum running speed; $p$—$p$ value; RPE—ratings of perceived exertion; s—seconds; sRPE—session ratings of perceived exertion; TD—total distance; TQR—total quality recovery; Wk—week; $\eta^2$—eta-squared.

The inter-week analysis presented significant differences in the TD ($p = 0.022$, $\Delta = 620.11$ m, $d = 0.39$). The sampled weekly microcycle showed a significant difference for the RPE and sRPE ($p = 0.002$, $\Delta = 0.76$ a.u., $d = 0.35$) and sRPE ($p = 0.002$, $\Delta = 67.90$ a.u., $d = 0.35$). The TQR score did not present significant differences in the inter-week analysis. Figure 3

presents the mean differences (%) between each weekly microcycle examined for external load, sRPE, RPE and TQR.

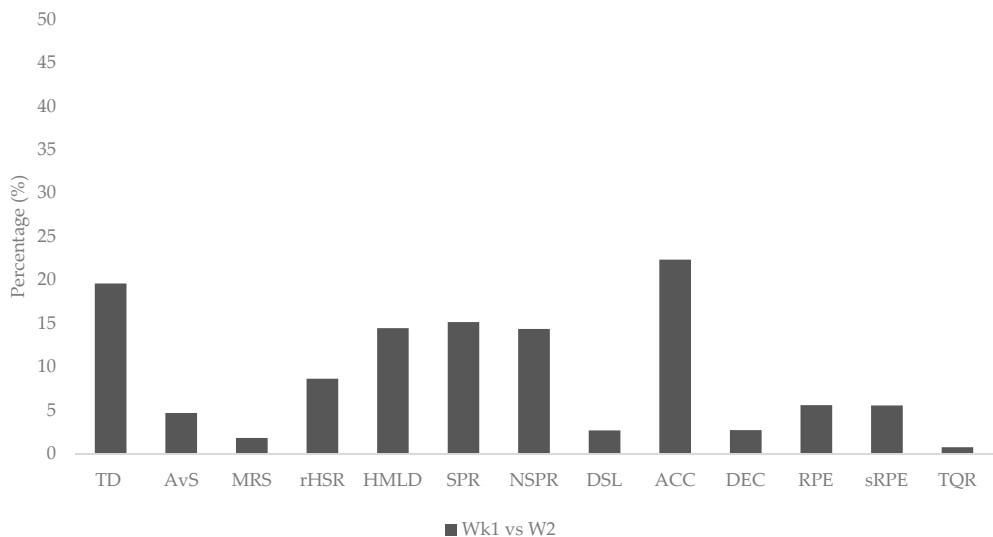

**Figure 3.** Mean training load and recovery status per session according to comparison between weekly microcycles.

### 3.4. Playing Position Analysis

Table 4 presents the descriptive statistics for mean weekly training load and recovery status for each playing position group analyzed.

**Table 4.** Mean training load and recovery status per session for each age group examined.

| | Variables | CB (*n* = 79) | FB (*n* = 65) | CM (*n* = 70) | WM (*n* = 62) | FW (*n* = 48) | *F* | *p* | η² | Post-Hoc |
|---|---|---|---|---|---|---|---|---|---|---|
| **External load** | TD (m) | 5282.28 ± 1407.51 | 5275.94 ± 1774.61 | 5456.91 ± 1565.86 | 5370.07 ± 1692.56 | 5156.90 ± 1820.92 | 0.20 | 0.037 | 0.003 | - |
| | AvS (m·min⁻¹) | 47.27 ± 13.98 | 51.15 ± 26.55 | 52.09 ± 23.08 | 52.16 ± 28.96 | 50.44 ± 25.53 | 0.47 | 0.758 | 0.007 | - |
| | MRS (m·s⁻¹) | 6.95 ± 1.04 | 7.29 ± 1.58 | 7.49 ± 1.54 | 7.28 ± 1.54 | 7.77 ± 3.77 | 1.36 | 0.246 | 0.18 | - |
| | rHSR (m) | 75.32 ± 71.00 | 66.29 ± 54.89 | 82.41 ± 74.30 | 71.73 ± 70.04 | 91.53 ± 110.93 | 0.30 | 0.018 | 0.037 | a |
| | HMLD (m) | 541.31 ± 243.65 | 548.51 ± 282.09 | 602.16 ± 275.41 | 562.16 ± 275.41 | 529.47 ± 360.56 | 0.88 | 0.475 | 0.012 | - |
| | Average sprint (m) | 44.29 ± 56.91 | 51.15 ± 26.55 | 49.06 ± 57.09 | 38.84 ± 48.85 | 56.58 ± 77.79 | 3.18 | 0.14 | 0.039 | a |
| | Number of sprints | 3.17 ± 3.30 | 2.69 ± 3.09 | 3.41 ± 3.63 | 3.08 ± 3.38 | 4.06 ± 5.14 | 1.02 | 0.400 | 0.013 | - |
| | DSL (a.u.) | 261.17 ± 141.37 | 230.52 ± 118.24 | 265.34 ± 149.59 | 238.11 ± 135.04 | 255.98 ± 123.91 | 0.73 | 0.573 | 0.010 | - |
| | ACC (m·s⁻²) | 44.63 ± 19.41 | 45.55 ± 20.04 | 47.77 ± 21.94 | 46.61 ± 24.01 | 45.25 ± 23.83 | 0.16 | 0.957 | 0.003 | - |
| | DEC (m·s⁻²) | 39.63 ± 18.71 | 40.22 ± 19.51 | 43.27 ± 22.19 | 41.18 ± 25.98 | 43.75 ± 34.44 | 0.30 | 0.875 | 0.005 | - |
| **Internal load** | RPE (a.u.) | 261.17 ± 141.37 | 230.52 ± 118.24 | 265.34 ± 149.59 | 238.11 ± 135.04 | 255.98 ± 123.91 | 2.89 | 0.023 | 0.034 | b |
| | sRPE (a.u.) | 44.63 ± 19.41 | 45.55 ± 20.04 | 47.71 ± 21.94 | 46.61 ± 24.01 | 45.25 ± 23.83 | 2.89 | 0.023 | 0.034 | b |
| **Recovery status** | TQR (a.u.) | 39.63 ± 18.71 | 40.22 ± 19.51 | 43.27 ± 22.19 | 41.18 ± 25.98 | 43.75 ± 34.44 | 1.28 | 0.279 | 0.016 | - |

Significant differences are verified as: (a) central defenders vs. forwards; (b) wide midfielders vs. forwards. Abbreviations: ACC—acceleration; a.u.—arbitrary unit; AvS—average speed; DEC—deceleration; *F*—F statistic; HMLD—high metabolic load distance; m—meters; min—minutes; MRS—maximum running speed; *p*—p value; RPE—ratings of perceived exertion; s—seconds; sRPE—session ratings of perceived exertion; TD—total distance; TQR—total quality recovery; η²—eta-squared.

The playing position comparison presented a significantly higher covered distance in rHSR (*p* = 0.037, Δ = 139.26 m, *d* = 0.17) and average sprint (*p* = 0.029, Δ = 142.13 m·s⁻¹, *d* = 0.18) for FW players than CD players. The internal training load measures presented a significant difference in playing position groups for RPE and sRPE. The RPE and sRPE was

higher in WM players than FW players ($p = 0.038$, $\Delta = 1.21$ a.u., $d = 0.065$). The TQR score did not present significant differences within playing position groups. Figure 4 presents the mean differences (%) between each playing position examined for external load, sRPE, RPE and TQR.

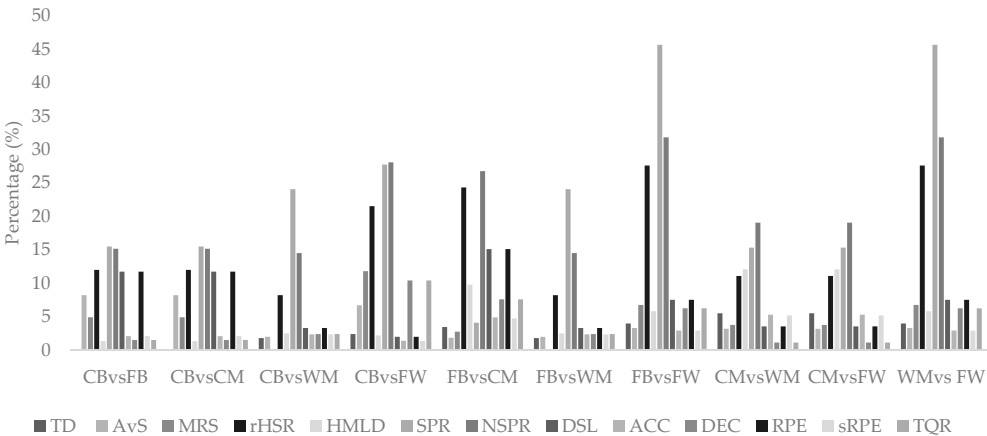

**Figure 4.** Mean training load and recovery status per session according to comparison between playing positions.

### 3.5. Interaction Effects between Age, Inter-Week, Inter-Day and Playing Position

The factorial ANOVA identified an interaction effect for TD between inter-day, inter-week and age ($F = 8.342$, $p = 0.008$). An interaction effect between the four groups analyzed was found for DEC ($F = 2.819$, $p = 0.017$).

## 4. Discussion

The aims of this study were to: (i) to quantify the weekly training load and recovery status of U15, U17 and U19 young players in a sub-elite football academy; and (ii) to analyze the influence of age, training day, weekly microcycle, training and playing position on the training load and recovery status. It was hypothesized that there are significant differences in weekly in-season training load according to age group, weekly microcycle, training day and positional role. Altogether, the findings of the present study describe significant differences between age group and training days. The inter-week analysis revealed differences only in the TD covered. The playing position had a minimal effect on the weekly training load. The recovery status presented only significant differences between age groups. For convenience, our findings were split into four analyses: (i) age group analysis; (ii) inter-day analysis; (iii) inter-week analysis; and (iv) inter-playing positions analysis.

### 4.1. Age Group Analysis

In the present study, all external training load measures presented a significant difference, except DSL (Table 1). Higher TD, AvS, MSR, rHSR, HMDL, average sprint and number of sprints were found in U17 players (moderate to strong effect). U17 players did not present significant differences when compared to U19 players in rHSR, average sprint distance, ACC and DEC. Our findings were consistent with Abade et al. [32], who reported a higher TD covered and average sprint distance for U17 players. The authors also did not find any differences between U17 and U19 players for body impact and relative impacts. In contrast, Wrigley et al. [33] reported a higher total weekly training load for older groups (i.e., U18). However, the authors measured physiological intensity, resorting to heart rate-based methods which could bias the high-intensity efforts [45,46,52]. Additionally, the data of the present study evidence the lowest intensity in U15 players' training sessions regarding rHSR, average sprint distance, number of sprints, ACC and DEC (strong effect). There was an age-related increase in the training intensity and to a greater extent

in the training volume [33,43]. Curiously, U15 players presented a higher TD and MSR when compared with U19 players (strong effect). There was an age-related increase in the training intensity and to a greater extent in the training volume [33]. Due to this fact, associated with a more conscious pacing strategy and better game interpretation with age, it was possible to also increase the exercise economics [53]. Even more interaction effects were found between inter-day, inter-week and age for TD, confirming an increase in the pacing strategy in the aging progression. Training periodization also seemed to influence the TD covered, concerning the training day and weekly microcycle.

The present investigation confirmed the age-related increase in the training intensity in sub-elite football. However, acceleration metrics have a high variability subsequent to the time-dependent and transient reductions [24,54]. Nonetheless, it was not clear from that acceleration whether variables increased across different age groups, G. Our study provides news insights about age influence in acceleration movements. We found an interaction effect between age, week, training day and playing position for DEC. In the current age group analysis, U19 players showed a higher internal training load (minimum to strong effect) and a lower recovery status (strong effect). These findings were consistent with Wrigley et al. [33], who noted a higher weekly RPE load in the older age group (i.e., U18). There was evidence that a higher training load and a good recovery were expected to improve physical performance [4,18]. Furthermore, we were able to verify a higher training volume in younger players (i.e., U15 vs. U19). It is reasonable to argue that coaching staff tend to code training programs with more volume and less intensity when it comes to younger players. Moreover, a focus on the basic tactical principles and technical skills using constrained training tasks was reported in younger age groups [32]. Nevertheless, the time spent at high-intensity zones and normalizing the session duration may affect the perceived exertion [14].

### 4.2. Inter-Day Analysis

The present inter-day analysis significant differences were presented in external training load for TD, MRS, AvS, rHSR, HMLD, average sprint distance, DSL and DEC (Table 2). Our training data show a lower TD covered in MD-1 (moderate to strong effect). Our main findings seem to be convergent in a strategy tapering based on a gradual reduction until the last day before MD. Normally, the training load pattern presents a progressive increase up to MD-3 and/or MD-4 followed by a decrease until MD-1 [21,54–56]. In what concerns youth football, the literature reported an unloading prior to the pre-match training session [19,31,56]. Our data show a higher AvS and MRS in MD-2. By contrast, MD-3 presented higher DSL and DEC. In the same line, previous studies reported a higher weekly training load in the middle of the week [31,56]. Moreover, a large variation has been reported between training days and within training sessions [19,56]. An unloading phase was normally adopted due to an optimal fitness recovery status for competition [57,58]. For instance, the high values found in the accelerometer variables during MD-1 could be used to monitor neuromuscular fatigue [59].

In the results of the present study, the internal training load and recovery status did not present significant differences for inter-day analysis. By contrast, previous studies reported an unloading phase in young players concerning RPE values [33,45]. Wrigley et al. [33] evidenced a tapering in U18 players. On the other hand, the U14 and U16 players presented relatively high training loads across the weekly microcycle. Indeed, it could be suggested that coaches opt for different tapering strategies when the age and competition focus increases.

### 4.3. Inter-Week Analysis

The external training load displayed inter-week differences in the TD covered (moderate effect) (Table 3). This indicates a small variation in the cumulative training load among weekly microcycles. Studies that assessed the seasonal loading were in line with the findings of the present study [20,21,23,60]. The literature reported a trivial increase in

seasonal training load [21–23]. A small increase in the training volume across seasonal phases was also reported [22,26].

The sampled RPE and sRPE showed significant differences among weekly microcycles (moderate effects). Our findings were in agreement with the literature that described a variation (i.e., 5–72%) in the weekly perceived load [19,20,56]. The authors deemed a long-term and systematic training monitoring to better understand the stage developments. The accumulated training load and the seasonal variation in the different stages of developments still constitute a research hot topic concerning youth football. Additionally, the sRPE provided consistent information about internal training throughout an entire season [15,61]. No significant differences were found in the TQR score for inter-week analysis. Previous studies were in the same line for perceived fatigue scores [19,20]. Selmi et al. [62] also showed that the perceived internal intensity was not influenced by the recovery state.

### 4.4. Inter-Playing Positions Analysis

The external and internal training loads were significantly different between playing position in the present study (Table 4). These differences were verified in rHSR and the average sprint distance between FW vs. CD players (minimum to moderate effect). Additionally, the internal training load presented significant differences between WM and FW players (minimum effect). The inter-positional variation was confirmed in a few studies addressing adult male and female professional football [23–25,40,46]. In youth football, the influence of playing position on physical and physiological performance during competition is well documented [27,63–68]. Although the positional role has been analyzed in training environments, the training load analysis was focused on constrained training tasks [29,30,69,70]. Therefore, it is important to know the influence of positional role in the weekly training load. Comparing elite and sub-elite football academies is also an important research gap which studies that examine accumulated weekly training load did not include in playing position analysis [15,31–33]. In the present study, the TQR score did not present significant differences within playing position groups. In contrast, the influence of playing position on the recovery status in the adult training football has been documented [20,71]. The results of this study seem to suggest that the influence of playing positions on the weekly training load and recovery status is unclear. A possible explanation could be related to the training tasks, which may not be representative of the positional role specificity [53]. However, future investigations may focus on high-demanding variations. Our findings show significant differences in two high-intensity variables (i.e., rHSR and average sprint distance). Additionally, the weekly training load quantification should consider the game model and representative game-based situations to promote playing position specificity [10–55].

Our study presented some limitations and the results should be interpreted with caution. First, the training data included only 2-week monitoring due to the constraints arising from the COVID-19 pandemic. Wherefore, the sample size was rather small, which reduces the understanding about training load and recovery variations across a competitive season. Additionally, the training data addressing only one sub-elite football academy cannot be extended to other teams and regions. This way, longitudinal samples should be considered in further investigations to examine seasonal variation. Second, biological maturation was not considered, which could bias the age group analysis [72,73]. Third, the current research analyzed overall training sessions rather than considering the training mode for each training exercise [29,30,40,46,69,70]. Integrating accumulated training load and matching data should be considered in further investigations on youth football. Indeed, there is a need for studies considering the effects of different week schedules (i.e., one-, two- and three-game week). The current literature analyzes the accumulated training load in professional football [74]. Furthermore, speed and acceleration thresholds in the present experimental approach were based on elite gold-standard guidelines. Future research should focus on the adjustment thresholds for elite and sub-elite youth football.

## 5. Practical Applications

This study points out new insights about accumulated weekly training and recovery status to sub-elite football players according to age group, training day, weekly microcycle and playing positions. Hence, there were several practical applications of the present training data. First, reporting accumulative training load could enhance the ecological validity more so than constrained tasks [32,75]. Second, the researchers and practitioners should consider age-related differences to design training programs. Similarly, periodization strategies must focus on a short- and mid-term implementation given the high intra-week variation and the low inter-week variation. Finally, the positional role did not seem to be a main factor affecting the weekly training load performed by young sub-elite football players.

## 6. Conclusions

This study allowed us to conclude that the weekly accumulated training load varied according to age group, training day, inter-week and playing position. There was an age-related influence of external training load (moderate to strong effect), internal training load (minimum to strong effect) and recovery status (strong effect). The external training load presented differences between training days (moderate to strong effect). External and internal training load were significantly different between playing positions. However, the playing position had a minimal effect on the weekly training load. This study provided specific insights about sub-elite youth football training load and recovery status to monitor training environments and load variations. Future research should include a longer monitoring period to assess training load and recovery variations across different season phases.

**Author Contributions:** Conceptualization, J.E.T. and P.F.; Data curation, J.E.T., J.R. and P.F.; Formal analysis, P.F., M.L. and R.F.; Funding acquisition, J.E.T., P.F. and A.M.M.; Investigation, J.E.T.; Methodology, P.F., T.M.B. and A.M.M.; Resources, P.F. and A.M.M.; Software, J.E.T., J.R. and M.L.; Supervision, A.J.S. and A.M.M.; Validation, P.F., T.M.B. and A.M.M.; Writing—original draft, J.E.T.; Writing—review and editing, P.F., R.F., A.J.S., T.M.B. and A.M.M. All authors have read and agreed to the published version of the manuscript.

**Funding:** This research was supported by the Douro Higher Institute of Educational Sciences and the Portuguese Foundation for Science and Technology, I.P. (project UIDB04045/2021).

**Institutional Review Board Statement:** The study was conducted according to the guidelines of the Declaration of Helsinki and approved by the institutional Ethical Committee from University of Trás-os-Montes e Alto Douro (Doc2-CE-UTAD-2021).

**Informed Consent Statement:** Informed consent was obtained from all subjects involved in the current investigation.

**Data Availability Statement:** Data are available under request to the contact author.

**Acknowledgments:** The authors express acknowledgement of all coaches and playing staff for cooperation during all collection procedures.

**Conflicts of Interest:** The authors declare no potential conflict of interest.

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
