# Peer review of "Quantifying Sub-Elite Youth Football Weekly Training Load and Recovery Variation"

_applsci, doi:10.3390/app11114871_

Round 1

Reviewer 1 Report

The work was aimed at analyzing the weekly in-season training load and recovery status by U15, U17 and U19 young players in a sub-elite football academy and comparing the weekly in-season training load and recovery status according age-group, weekly microcycle, training day and positional role. It is interesting work but need some serious improvements especially regarding statistical analysis. There are some comments I’d like to express:

  • As the Authors state themselves, the 2-week-long observation may be too short especially that it does not take the cumulative fatigue into account. Therefore, these observations repeated in the middle and at the end of the competitive season would provide interesting data. I’m aware that it could be hardly possible taking the COVID-19 pandemic into account. However, this is my suggestion for the Authors for the future research.
  • The accuracy of the participants’ age is exaggerated. For example, the age of 13.28 equals ca. 4847.2 days (365 × 13.28). Do really the authors have such precise information about the age of the players (including parts of the days)? I can understand that it is a mathematical operation result. However, one must remember that output cannot be more precise that input data.
  • Similarly, the statement in the lines 138-139 “The training sessions was on average 18.2±0.4 players.” should be corrected informing that that there were on average 18 players.
  • Accelerations and decelerations are measured as m×s-2 instead of m×s2 (line 174).
  • Why ACC and DEC in the Tables 2-5 are measured as m×s-1?
  • The citation style in line 214 should be changed to be number instead of providing cited author’s name. It is also missing in the References
  • If the Authors compare analyzed variables as age-, inter-day-, inter-week- and inter-playing positions-related, they should perform factorial ANOVA (age × day × week × playing position). For example, they provide evidence that the variables significantly differ in the age groups. The question therefore is – how does it influence the overall age analysis in all other analyses?
  • Although I don't feel qualified to judge about the English language and style, there are typographic mistakes and repetitions that should be removed (e.g. lines 31-32, 227 etc.).
  • Data Availability Statement section should be provided.

Author Response

Thank you very much for the time you spent and your feedback on this manuscript. Please, see attachment.

Reviewer 2 Report

Please see word document attached.

Author Response

(The authors gave the same response as above.)

Reviewer 3 Report

This investigation quantified training load and recovery status in sixty youth soccer players. In general, this research provides novel data and the protocols were well-conducted. 

General Comments: 

It is clear that the paper has undergone one round of revisions prior to my review. Based on an examination of the changes from the initial manuscript, I have no further methodological suggestions. 

However, it is readily apparent that this manuscript requires extensive English language editing, as there are numerous areas of the text that are unclear to the reader. There are a huge amount of grammar, syntax, pluralization, and nomenclature errors present throughout the body of the manuscript that must be corrected prior to publication. Please correct these issues to ensure the reader is able to follow along with the work being described in the manuscript. 

Similarly, I think that the title of this manuscript should be changed, as it is somewhat grammatically-incorrect. I recommend changing the title to something along the lines of "Quantifying sub-elite youth football training load and recovery variations across a competitive season"

Author Response

(The authors gave the same response as above.)

Round 2

Reviewer 1 Report

The work has been improved by the Authors and I have only a few minor comments:

  • The Authors did not understand my suggestion regarding the accuracy of the participants’ age. I’d like to point out that providing age with accuracy up to second decimal number (e.g. 13.28 years) not having the precise information about the age of the players (including parts of the days), as stated by the Authors is not appropriate.
  • It was necessary to exclude Table 1 but it is Authors’ decision how to describe the study group.
  • Table 5 in revised version of the manuscript should be re-numbered as Table 4.

Author Response

(The authors gave the same response as above.)

Reviewer 2 Report

I would like to thank the reviewers for their detailed efforts in responding to my comments however, I do not not feel this manuscript is of an appropriate level for publication based upon the following observations.

1.     It is still not clear what question this study is attempting to answer; for example, the title suggests the study is an observation across a competitive season, line 108, states only 2-weeks data collection were achieved, Line 112 contradicts this by stating training load was continuously monitored for a full month.

2.     If this is a two-week data collection how can the authors justify the comparisons between week 1 and week 2, surly training loads naturally vary across a training cycle? What is the theoretical basis of these comparisons?

3.     I feel the two-week data collection period is also a serious limitation to comparisons between age groups and by position. How do we know that the differences in external or internal load are not merely differences in coaching emphasis or natural variation?

4.     It feels like this study is attempting to answer multiple questions with a limited data set: line 18 (abstract) states aim “2” is to investigate the effect of 4 different independent variables on 3 different dependent variables (with multiple metrics – e.g. external load).

There are a number of other more minor concerns which include the presentation of results which are still confusing, particularly the figures that convert data into percentages. A clear presentation of the magnitude and uncertainty of the differences between groups would be beneficial. Again, the broad aims do not help the data presentation here.

I am not a statistician (and happy to rebutted) but my limited knowledge here makes me question the analysis methods at least for some of the questions. For example, when comparing between age groups within and between player variation needs to be accounted for.

There are still multiple examples of poor spelling grammar that unfortunately make the manuscript difficult to read in places.

Author Response

(The authors gave the same response as above.)
